# Human Muse cells reduce myocardial infarct size and improve cardiac function without causing arrythmias in a swine model of acute myocardial infarction

Yoshihisa Yamada[1], Shingo Minatoguchi[1], Shinya Baba[2], Sanae Shibata[3], Satoshi Takashima[3], Shohei Wakao[4], Hiroyuki Okura[1], Mari Dezawa[4], Shinya Minatoguchi[2,5]*

1 Department of Cardiology, Gifu University Graduate School of Medicine, Gifu, Japan, 2 Gifu Municipal Hospital, Gifu, Japan, 3 Animal Teaching Hospital (Anesthesiology) Faculty of Applied Biological Science, Gifu University, Gifu, Japan, 4 Department of Stem Cell Biology and Histology, Tohoku University Graduate School of Medicine, Sendai, Miyagi, Japan, 5 Department of Circulatory and Respiratory Advanced Medicine, Gifu University Graduate School of Medicine, Gifu, Japan

* minatos@gifu-u.ac.jp

**Data Availability Statement:** All relevant data are within the paper and the Supporting information files.

## Abstract

### Background

We recently reported that multilineage-differentiating stress enduring (Muse) cells intravenously administered after acute myocardial infarction (AMI), selectively engrafted to the infarct area, spontaneously differentiated into cardiomyocytes and vessels, reduced the infarct size, improved the left ventricular (LV) function and remodeling in rabbits. We aimed to clarify the efficiency of Muse cells in a larger animal AMI model of mini-pigs using a semi-clinical grade human Muse cell product.

### Method and result

Mini-pigs underwent 30 min of coronary artery occlusion followed by 2 weeks of reperfusion. Semi-clinical grade human Muse cell product ($1 \times 10^7$, Muse group, n = 5) or saline (Vehicle group, n = 7) were intravenously administered at 24 h after reperfusion. The infarct size, LV function and remodeling were evaluated by echocardiography. Arrhythmias were evaluated by an implantable loop recorder. The infarct size was significantly smaller in the Muse group (10.5±3.3%) than in the Vehicle group (21.0±2.0%). Both the LV ejection fraction and fractional shortening were significantly greater in the Muse group than in the Vehicle group. The LV end-systolic and end-diastolic dimensions were significantly smaller in the Muse group than in the Vehicle group. Human Muse cells homed into the infarct border area and expressed cardiac troponin I and vascular endothelial CD31. No arrhythmias and no blood test abnormality were observed.

**Funding:** This study was supported by a grant-in-aid from the Japan Agency for Medical Research and Development, and JSPS KAKENHI Grant Number JP20K08400, Japan. The funder had no role in study design, data collection and analysis, decision to publish, or preparation of the manuscript. This was newly included in the main text.

**Competing interests:** The authors have declared that no competing interests exist.

## Conclusion

Muse cell product might be promising for AMI therapy based on the efficiency and safety in a mini-pig AMI.

## Introduction

Stem cell therapies are hopeful treatments for acute myocardial infarction (AMI). It has been reported that intravenous injection of bone marrow (BM)-derived mononuclear cells (MNCs) led to a reduction in infarct size and recovery of cardiac function in AMI models of rabbit and pigs [1, 2] and also that intravenous injection of BM-derived mesenchymal stem cells (MSCs) led to a reduction in infarct size and recovery of cardiac function in AMI models of rats and pigs [3, 4]. However, clinical trials using BM-derived MNCs and MSCs after AMI demonstrated minimal improvement of cardiac functions, if any [5, 6]. Therefore, more powerful stem cell alternatives are required.

Multilineage-differentiating stress enduring (Muse) cells, able to differentiate into endodermal, ectodermal, and mesodermal cells without tumorigenicity, were identified in the BM, peripheral blood and organ connective tissues, as cells positive for pluripotent surface marker, stage-specific embryonic antigen (SSEA)-3 [7–12].

We recently reported in a rabbit AMI model that intravenously administered autograft and allograft Muse cells, which express HLA-G to exhibit immunotolerance, specifically engrafted to the infarct area via S1P-S1PR2 axis, reduced the infarct size, improved the cardiac function and remodeling through spontaneous differentiation into cardiomyocytes as well as paracrine effects [13]. However, the efficiency and safety of semi-clinical grade human Muse cell preparation in a larger animal AMI model has not yet been investigated. Therefore, we examined the effects of human Muse cell product on the infarct size, cardiac function and remodeling, arrhythmias, and blood test in a mini-pig model of AMI.

## Methods

### Preparation of human Muse cells

Semi-clinical grade human Muse cell preparation was provided by Clio, Inc. (merged into Life Science Institute, Inc.Tokyo). Human Muse cell–based product was produced from human mesenchymal stem cells (MSCs) after exposing the cells to the combination stresses. We prepared human green fluorescence protein (GFP)-Muse cells [3, 5] from GFP-introduced BM-MSCs as SSEA-3 (+) cells, as previously described [13].

### Animal model and protocol

Animals received humane care in accordance with the Guide for the Care and Use of Laboratory Animals (NIH publication 85–23, revised 1996). The protocol was approved by the Ethical Committee of Gifu University Graduate School of Medicine (permission number: 28–7). The experiment was performed according to the ARRIVE Guidelines (https://www.nc3rs.org.uk/arrive-guidelines).

The mini-pigs (Nippon Institute for Biological Science: 6.4 ~ 12.4 kg, average: 9.4 kg) were anesthetized by propofol (2 mg/kg IV) and maintained with sevoflurane in oxygen under mechanical ventilation (KV-1a, Kimura Ikakiki Co., Tokyo, Japan). Analgesia was maintained by fentanyl (20 μg/kg/hr, IV). Surgical procedures were performed aseptically. The arterial

pressure during the course of the experiment was measured using an arterial pressure monitor from the peripheral artery. Thereafter, the animals were systemically heparinized (500 U/kg). After a left thoracotomy was performed in the fourth intercostal space, the heart was exposed and 4–0 silk thread was placed beneath the left anterior descending coronary arterial branch coursing down the middle of the anterolateral surface of the left ventricle. Both ends of the silk suture were then passed through a small vinyl tube, and the coronary branch was occluded by pulling the snare, which was fixed by clamping the tube with a mosquito hemostat. The coronary artery was occluded for 30 min and reperfused to create an AMI model. Myocardial ischemia was induced for 30 min. Myocardial ischemia was confirmed by ST-segment elevation on the electrocardiogram and regional cyanosis of the myocardial surface. Reperfusion was confirmed by myocardial blush over the risk area after releasing the snare, as described previously [13].

For the assessment of effects on the infarct size reduction and function recovery, animals received intravenous injection of $1 \times 10^7$ semi-clinical grade human Muse cell preparation (2 mL) (Muse group, n = 5) or 2 mL of saline (Vehicle group, n = 7) at 24 hours after reperfusion of occluded coronary artery for 30 min without immunosuppressants and observed for 2 weeks. For the assessment of Muse cell differentiation into cardiomyocytes and vessels in histologic analysis, additional AMI animals (n = 3) received intravenous injection of $1 \times 10^6$ of GFP-labeled-human Muse cells [13]. The investigators evaluating the outcomes were blinded to the protocol.

## Allocation of the animals

Totally 20 minipigs were initially allocated to Vehicle group (n = 10) or Muse cells group (n = 10) alternately before we made AMI models. However, 6 animals died because of ventricular fibrillation and ventricular rapture during the coronary occlusion and immediately after reperfusion before the administration of vehicle or Muse cell product (2 allocated to Vehicle group and 4 allocated to Muse group). Therefore, finally, 6 Muse and 8 Vehicle were intravenously administered at 24 h after AMI. Out of them, 1 Muse minipig died 1 day after and 1 Vehicle died 3 days after. Finally, 5 Muse-minipigs and 7 Vehicles were survived for 14 days.

## Echocardiography

Echocardiography (SSD2000, Aloka Co., Ltd.) was performed at 2 weeks after AMI. The left ventricular (LV) ejection fraction (EF), fractional shortening (FS), LV end-systolic dimension (LVESd), and LV end-diastolic dimension (LVEDd) were obtained. EF was measured by Teichholz method using M-mode echocardiography.

## Myocardial infarct size

Animals were sacrificed with an overdose of pentobarbital and KCL at 2 weeks after AMI. The heart was dissected out and LV was sectioned into seven transverse slices parallel to the atrioventricular ring. Each slice was fixed with 10% buffered formalin for 4 h, embedded in paraffin, and cut into 4-µm-thick sections. Transverse LV slices at the papillary muscle level were stained with Masson-Trichrome. The LV areas and infarct areas ($mm^2$/slice) were calculated using an image analyzer (Win ROOF, version 7.4, Mitani Corporation, Tokyo, Japan) connected to a light microscope (BZ-8000, KEYENCE, Osaka). The infarct size is expressed as a percentage of the LV. Since it is generally accepted that the Evans blue dye/TTC method is not reliable for evaluating infarct size after 72 hours reperfusion because of remodeling due to scar shrinkage within the infarct [14], we used Masson-Trichrome staining to assess the infarct size as percentage of LV.

## Laser confocal microscopic observations for cardiac markers

Mini-pigs receiving GFP-labeled human Muse cells were sacrificed with an overdose of pento-barbital and KCL at 2 weeks. The cardiac tissue was fixed with 4% paraformaldehyde in PBS. Cryosections (8 μm thick) were cut and stained as previously described [13]. Primary antibodies used were: anti-cardiac troponin I (1:20; Abcam), anti-CD31 (1:20; Abcam), and anti-GFP (1:1000, Abcam). The secondary antibodies used were either anti-Chicken IgY H&L (Alexa Flour 488, Abcam), anti-Rabbit IgG H&L (Alexa Flour 680, Abcam), or anti-mouse IgG H&L (Alexa Flour 647, Abcam). Images were captured using a confocal microscope (C2si; Nikon Tokyo, Japan).

## Immunohistochemistry for CD31

Paraffin sections were stained with anti-CD31 (1:100; Dako) followed by HRP-conjugated anti-rabbit IgG and detected by the HRP-3,3'-diaminobenzidine system (Wako). Images were captured using a light microscope (BX 50 F4, OLYMPUS, Tokyo).

## Blood test

Peripheral blood count and blood biochemical test were performed before and 2 weeks after AMI.

## Assessment of arrythmias

In some of the Muse group (n = 3) and Vehicle group (n = 1), an implantable loop recorder (Reveal XT, Medtronic PLC, Minneapolis, USA) was placed in the 4th intercostal subcutaneous tissue where R waves could be recorded by electrocardiogram when the chest was closed, and arrhythmias were evaluated for 2 weeks after AMI. In this instrument, heart rate more than 161/min was set to be recorded as tachycardia.

## Statistical analysis

Values are expressed as the mean ± standard error. Differences between 2 groups were assessed by paired or unpaired Student's-t test (Stat View, J5.0 software, HULINKS Inc.). Values of $p < 0.05$ were considered significant, and values of $p < 0.01$ and $p < 0.001$ were considered highly significant.

# Results

## Physiological findings

Fig 1A shows blood pressure and heart rate. There was no significant difference in systolic blood pressure, diastolic blood pressure or heart rate between the two groups. Fig 1B shows a typical case of cardiac echocardiography in Vehicle and Muse groups. The ejection fraction (EF) was significantly greater in the Muse group (76.1± 0.6%) than in the Vehicle group (58.3± 2.7%) at 2 weeks in Fig 1C. Fractional shortening (FS) was significantly greater in the Muse group (42.0± 0.6%) than in the Vehicle group (29.3±1.9%) at 2 weeks in Fig 1C. The LV end-systolic dimension (LVESd) was significantly reduced in the Muse group as compared with the Vehicle group (Muse vs. Vehicle: 11.0±1.0 vs. 16.6± 0.6 mm, p = 0.0006) in Fig 1C. The LV end-diastolic dimension (LVEDd) was also reduced in the Muse group compared with the Vehicle group (Muse vs. Vehicle: 19.1±1.8 vs. 23.6±1.0 mm, p = 0.04) in Fig 1C.

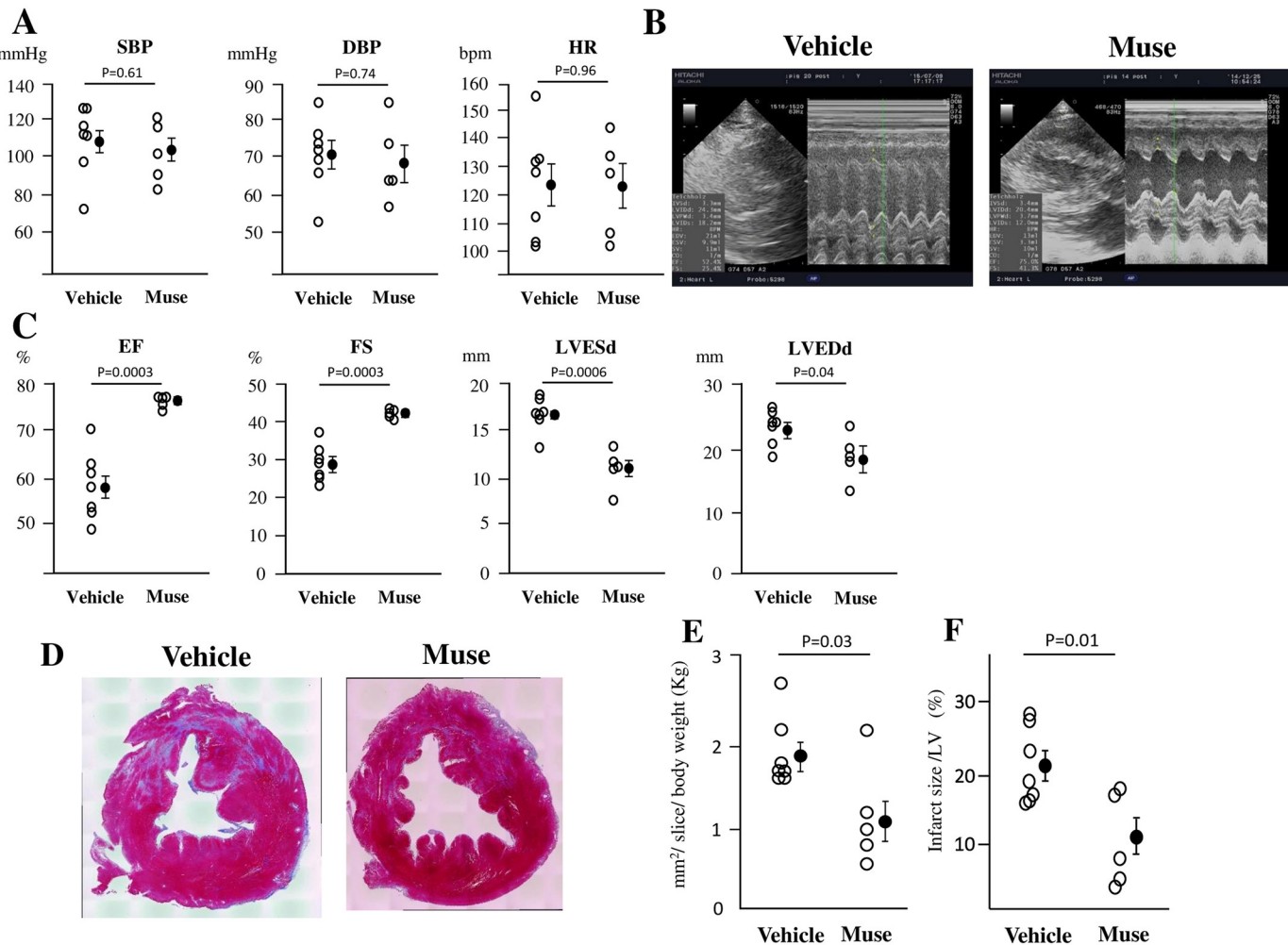

**Fig 1. Cardiac function and myocardial infarct size.** A: Systolic blood pressure (SBP), diastolic blood pressure (DBP) and heart rate (HR) in the Vehicle (n = 7) and Muse (n = 5) groups. B: Representative echocardiography in Vehicle and Muse groups. C: Left ventricular ejection fraction (LVEF), LV fractional shortening (LVFS), LV end-diastolic dimension (LVEDd), and LV end-systolic dimension (LVESd) in the Vehicle (n = 7) and Muse (n = 5) groups. D: Representative pictures of the Masson-Trichrome staining of cross-sections of LV in the Vehicle and Muse groups. Blue color-coded areas indicate the infarct regions and red-coded areas indicate the non-infarct regions of LV. E: Fibrosis area (mm$^2$/slice/kg) in the Vehicle (n = 7) and Muse (n = 5) groups. F: Infarct size as a percentage of LV area (%) in the Vehicle (n = 7) and Muse (n = 5) groups.

## Infarct size

Fig 1D shows representative pictures of Masson-Trichrome staining of cross-sections at the papillary muscle level of the LV. The infarct area corrected by slice/body weight kg (mm$^2$/slice/kg) was significantly smaller (p = 0.03) in the Muse group (1.1± 0.2 mm$^2$/slice/kg) than in the Vehicle group (1.9± 0.1 mm$^2$/slice/kg) in Fig 1E. The infarct size as a percentage of the LV was significantly smaller (p = 0.01) in the Muse group (10.5± 3.3%) than in the Vehicle group (21.0±2.0%) in Fig 1F.

## Immunohistochemistry

Using fluorescent staining method, GFP-labeled Muse cells expressed cardiac troponin I, a cardiomyocyte marker, in the infarct border area in the Muse group in Fig 2A. Fig 2B is a

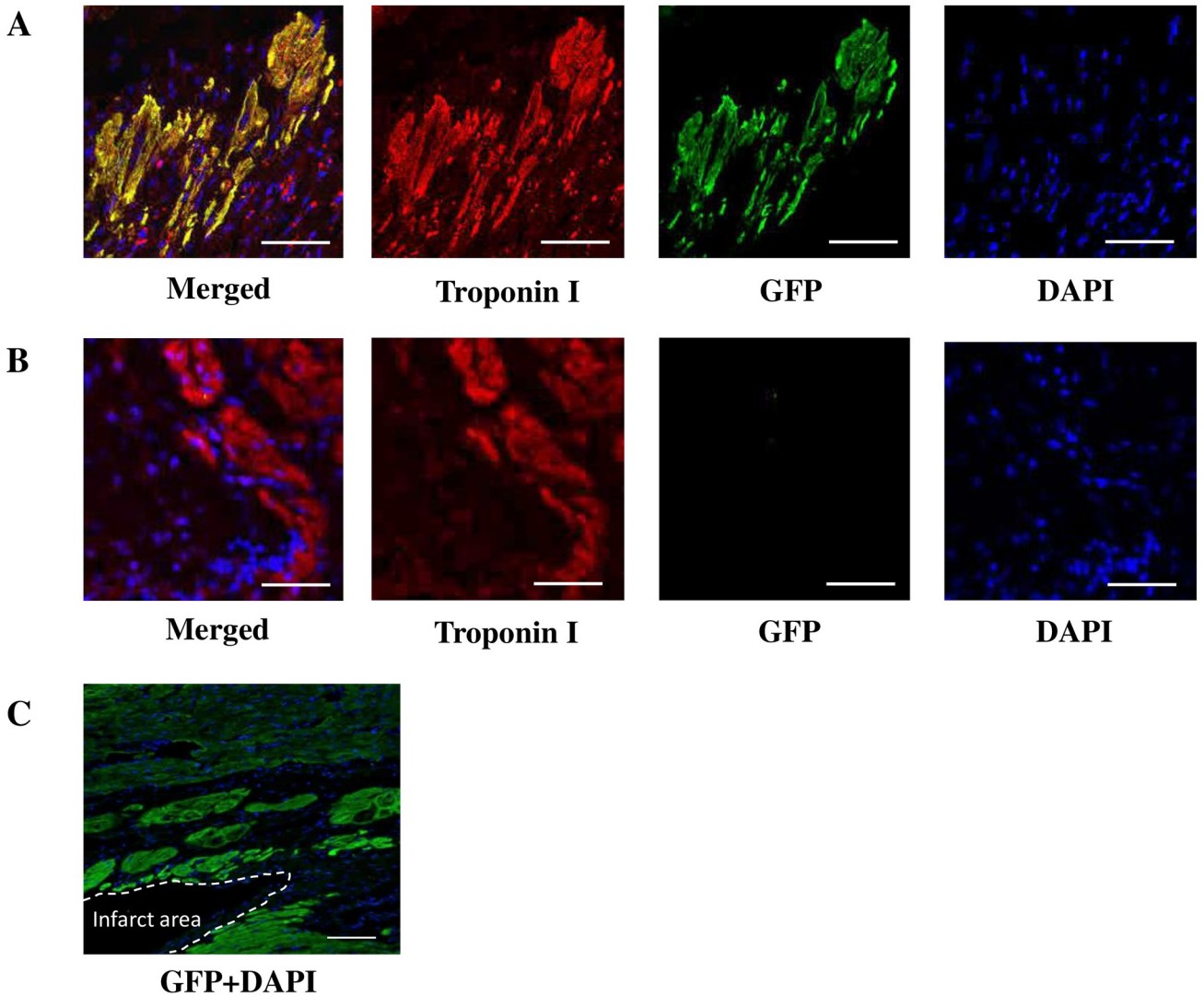

**Fig 2. Differentiation of Muse cells into cardiomyocytes.** A: GFP-labelled Muse cells and cardiac troponin I in the infarct border area. GFP (green) and troponin I (Red) are merged, suggesting that GFP-labelled Muse cells differentiated into cardiomyocytes. Bar = 50 μm. B: Negative control in the infarct border area. Troponin I positive cells (red), DAPI (blue), Bar = 50 μm. C: GFP (green)-labelled Muse cells are engrafted in the infarct border area. Bar = 100 μm.

negative control in the vehicle group. The GFP-labeled Muse cells were mainly detected in the infarct border area of the myocardium in Fig 2C. GFP-labeled Muse cells expressed cardiac CD31, a vascular endothelial cell marker, in the infarct border area in the Muse group in Fig 3A. Fig 3B is a negative control in the vehicle group. The number of CD31-positive vessels by immunohistological staining was significantly greater (p = 0.002) in the Muse group than in the vehicle group in Fig 3C and 3D.

## Blood test

No abnormality in the blood cell count or biochemical test was found in either group in Table 1.

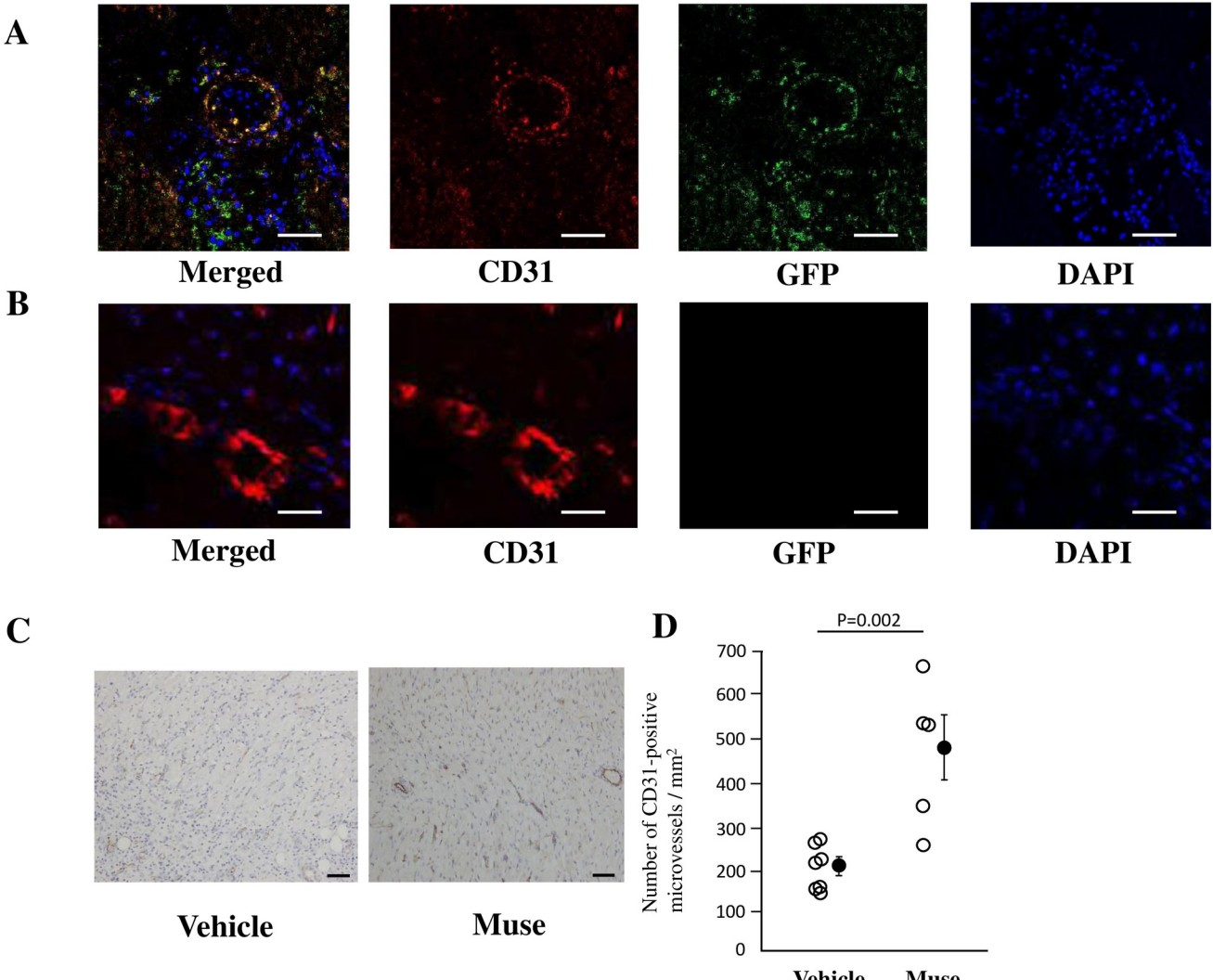

**Fig 3. Differentiation of Muse cells into vessels.** A: GFP-labelled Muse cells and CD31-positive microvessels in the infarct border area. GFP (green) and CD31 (Red) are merged, suggesting that GFP-labelled Muse cells differentiated into vascular endothelium. Bar = 50 μm. B: Negative control in the infarct border area. CD31 (red), DAPI (blue), Bar = 25 μm. C: Typical pictures of CD31-positive microvessels by immunohistological staining in the Vehicle and Muse groups. Bars = 50 μm. D: The number of CD31-positive microvessels by immunohistological staining is significantly greater in the Muse (n = 5) group than in the Vehicle (n = 7) group.

## Effect of Muse cells on arrhythmias

Fig 4 shows that analysis of the implantable loop recorder demonstrated that no arrhythmias such as tachycardia, bradycardia, pause, ventricular and supraventricular extra-systoles, ventricular tachycardia, or ventricular fibrillation were observed during the 2 weeks in the Muse group (n = 3) or in the Vehicle group (n = 1).

## Discussion

We recently reported that intravenously administered Muse cells significantly reduce the myocardial infarct size, improve the LV function, and attenuate LV remodeling in a rabbit model of AMI [13]. The administered Muse cells engrafted to the infarct and infarct border areas of

**Table 1. Blood cell count and biochemical data.**

| | Vehicle (n = 7) | | Muse (n = 5) | |
|---|---|---|---|---|
| | before | after | before | after |
| WBC ($10^3$/μL) | 9257 ± 964 | 10071 ± 698 | 9020 ± 1240 | 8840 ± 1144 |
| RBC ($10^4$/μL) | 686 ± 27 | 684 ± 31 | 697 ± 54 | 695 ± 60 |
| Hb (g/dL) | 10.5 ± 0.4 | 11.0 ± 0.7 | 11.8 ± 0.9 | 12 ± 1.1 |
| AST (U/L) | 35 ± 2.8 | 40 ± 2.7 | 43 ± 7.6 | 46 ± 4.5 |
| ALT (U/L) | 33 ± 1.6 | 38 ± 2.5 | 41 ± 2.4 | 40 ± 3.0 |
| BUN (mg/dL) | 3.9 ± 1.0 | 5.2 ± 1.2 | 4.1 ± 1.3 | 2.8 ± 0.5 |
| Cre (mg/dL) | 0.5 ± 0.1 | 0.6 ± 0.1 | 0.4 ± 0.1 | 0.4 ± 0.1 |
| Na (mEq/L) | 140 ± 1.3 | 145 ± 1.7 | 142 ± 0.4 | 143 ± 0.5 |
| K (mEq/L) | 4.8 ± 0.2 | 5.3 ± 0.2 | 4.8 ± 0.2 | 4.9 ± 0.1 |
| Cl (mEq/L) | 102 ± 2.0 | 105 ± 1.8 | 103 ± 0.9 | 103 ± 0.4 |

the heart at a high rate of ∼14% of the injected Muse cells, which was mediated through the S1P-S1PR2 axis; an interaction between sphingosine-1-phosphate (S1P) produced in the damaged heart and S1P receptor 2 (S1PR2) located on Muse cells [13]. The engrafted autograft GFP-labeled Muse cells expressed the cardiac markers ANP, troponin I, and α-actinin at 2 weeks and 2 months after AMI, and expressed the vascular endothelial marker CD31 and vascular smooth muscle marker α-smooth muscle actin at 2 weeks, suggesting that Muse cells

# Arrhythmia analysis using an implantable loop recorder

**A** Implantable loop recorder
(Medtronic PLC)

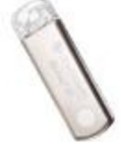

**B** Episode summary for 2 weeks in a case of Muse groups

| EGM | | | 📹 |
|---|---|---|---|
| tachycardia | bradycardia | pause | Patient record |
| 37 | 0 | 0 | 0 |

**C** Recorded electrocardiogram (ECG) showing sinus rhythm with 150/min of heart rate

A typical case of sinus rhythm

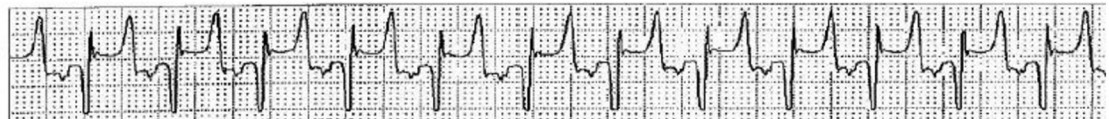

**Fig 4. Analysis of arrythmias using an implantable loop recorder.** A: Implantable loop recorder. B, C: Episode summary for 2 weeks in a case of Muse groups, showing no arrythmias such as bradycardia, pause except for tachycardia. we see the recorded electrocardiogram as shown in C, this was judged as a 300/min of tachycardia in the loop recorder. However, careful analysis of electrocardiogram revealed that the recorder judged T wave and R wave as a one beat and then the heart rate was regarded as a 300/min of tachycardia, but it was actually a 150/min of heart rate. Therefore, this was not a real tachycardia of 300/min.

differentiated into cardiomyocytes and vessels. Cardiomyocytes differentiated from G-CAM-P3-labelled Muse cells expressed the gap junction marker connexin 43 and exhibited G-CaMP3 activity synchronous with the heartbeat of systole and diastole, suggesting that Muse cells differentiated into working cardiomyocytes with physiologic activity and normal electromechanical integration between the graft and host [13].

On the basis of the experiment on the effects of Muse cells in a rabbit model of AMI [13], in the present study, we performed an experiment on the effects of human Muse cells in a larger animal mini-pig model of AMI as the next step for clinical trials. In this study, in consistent with our previous report in a rabbit model of AMI [13], semi-clinical grade human Muse cell preparation markedly reduced the infarct size, improved the LV function, and attenuated LV remodeling in a mini-pig model of AMI in Fig 1. Muse cells engrafted to the heart and expressed cardiac troponin-I, suggesting that they differentiated into cardiomyocyte lineages. The number of CD31-positive microvessels in the infarct border area was significantly greater in the Muse group than in the Vehicle group. Muse cells labelled with GFP differentiated into CD31-positive microvessels, suggesting that they differentiated into vascular endothelial cells. Therefore, Muse cells contributed to neovascularization, which is essential for tissue repair and maintenance of reconstructed tissue.

The arrhythmogenicity is caused by re-entrant pathways due to heterogeneity in conduction velocities between the graft and host [15]. As a matter of fact, it has been reported that myoblast from a skeletal muscle injected into myocardium in patients with depressed LV function caused arrythmias such as ventricular tachycardia [16]. However, in the present study, the implantable loop recorder detected no arrhythmias during 2 weeks after the administration of semi-clinical grade human Muse cell preparation, suggesting that Muse cell treatment does not cause any arrythmias. This may be related to the previous report that engrafted Muse cells to the infarcted heart differentiated into working cardiomyocytes with physiologic activity and normal electromechanical integration between the graft and host [13].

In addition, the peripheral blood cell count or blood biochemical test did not show any abnormality.

These results suggest that human Muse cell product reduce infarct size and improve cardiac function without causing arrhythmia and abnormality in peripheral blood cell count or blood biochemical data in a swine model of acute myocardial infarction.

## Study limitation and clinical perspective

In the present study, we did not calculate the total number of engrafted GFP-labelled Muse cells in the heart and the differentiation rate of Muse cells into cardiac troponin I-positive cells because the heart of the swine is too big to calculate the whole number of engraftment and differentiation. However, we have previously estimated the total number of integrated GFP-labelled Muse cells in the AMI heart of the rabbit whole heart, and the number of engrafted Muse cells was 43,555±11992 cells corresponding to 14.5±4.0% of the total number of injected 300,000 of Muse cells [13], and the differentiation rate of Muse cells into cardiac markers cardiac troponin I was 14.4±3.3% at 2 weeks after AMI in rabbits [13].

Because the evaluation of arrythmias by loop-recorder was performed in the small number of Muse group (n = 3) and Vehicle group (n = 1), larger number of arrythmia evaluation is warranted.

However, on the basis of the results of animal studies using rabbits [13] and mini-pigs in the present study, we have already performed a first-in-human clinical trial using an allogenic human Muse cell-based product, CL2020, for the treatment of patients with acute myocardial infarction after completion of the necessary process under approval of regulatory authorities.

As a result, CL2020 was safe and significantly improved cardiac function at 12 weeks after the onset of AMI [17]. We confirmed in this first-in-human clinical trial that administration of Muse cell product did not affect biological test such as LDH, ALT, AST, ALP, r-GTP, total bilirubin, BUN and creatinine, and inflammatory cytokines such as IL-1β, TNF-α, IL-6 and INF-γ during 12 weeks [17]. On the basis of this result, a randomized, double-blinded, placebo-controlled, multicenter clinical trial to examine the effects of an allogenic human Muse cell-based product, CL2020 on LV function and LV remodeling and safety in patients with AMI is currently in progress. We previously reported that when the effect on the infarct size and cardiac function were compared between Muse cells and mesenchymal stem cells (MSCs), both the reduction in the infarct size and improvement of cardiac function were significantly greater in the Muse group than in the MSC group [13]. This may support that Muse cells therapy for AMI has potential ability to show the clinical benefit in the clinical trial, although many clinical trials using somatic stem cells including bone marrow mononuclear cells and MSCs failed to show the clinical benefit in the clinical trials [5, 6].

Muse cells can be isolated as SSEA-3(+) cells from various sources since they normally reside in the bone marrow ($\sim$0.03% of mononucleated cell population), peripheral blood and organ connective tissue [7, 18]. However, practical sources will be the bone marrow, adipose tissue, dermal fibroblasts and umbilical cord. Muse cells are also collectable from commercially released MSCs and fibroblasts, and are contained as several percent of these cultured cells [7, 18]. Therefore, Muse cells are accessible.

The doubling time of Muse cells is $\sim$1.3 days/cell division [7, 18]. This is nearly the same or slightly longer than that of human fibroblasts. Since they are not tumorigenic, unlike ES and iPS cells, they do not show exponential proliferative activity. Nevertheless, they are expandable to a clinical scale. Indeed, Life Science Institute Inc., a group company of Mitsubishi Chemical Holdings Corporation, succeeded in producing clinical grade Muse cell formula, CL2020. CL2020 is already applied to clinical trials [17].

In conclusion, human Muse cell product may be promising for AMI treatment based on efficiency and safety in a mini-pig model of AMI.

## Supporting information

**S1 File.**
(DOCX)

**S1 Data.**
(XLSX)

## Acknowledgments

We thank Mrs Noriko Endo for her technical assistance.

## Author Contributions

**Conceptualization:** Yoshihisa Yamada, Shinya Minatoguchi.

**Data curation:** Yoshihisa Yamada, Shingo Minatoguchi, Shinya Baba, Sanae Shibata, Satoshi Takashima, Shohei Wakao, Hiroyuki Okura, Mari Dezawa, Shinya Minatoguchi.

**Formal analysis:** Yoshihisa Yamada, Shinya Minatoguchi.

**Funding acquisition:** Yoshihisa Yamada, Shinya Minatoguchi.

**Investigation:** Shinya Minatoguchi.

**Methodology:** Shinya Minatoguchi.

**Project administration:** Shinya Minatoguchi.

**Resources:** Shinya Minatoguchi.

**Software:** Shinya Minatoguchi.

**Supervision:** Shinya Minatoguchi.

**Validation:** Shinya Minatoguchi.

**Visualization:** Shinya Minatoguchi.

**Writing – original draft:** Yoshihisa Yamada, Mari Dezawa, Shinya Minatoguchi.

**Writing – review & editing:** Shinya Minatoguchi.

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
