## [Decision Letter · Decision Letter 0]

22 Dec 2021

PONE-D-21-37007Human Muse cells reduce myocardial infarct size and improve cardiac function without causing arrythmias in a swine model of acute myocardial infarctionPLOS ONE

Dear Dr. Minatoguchi,

Thank you for submitting your manuscript to PLOS ONE. After careful consideration, we feel that it has merit but does not fully meet PLOS ONE’s publication criteria as it currently stands. Therefore, we invite you to submit a revised version of the manuscript that addresses the points raised during the review process.Please discuss why muse cells are superior to MSCs.With respect to clinical use, are the muse cells easy to isolate and expand to a large amount in vitro?Please adequately address reviewers' comments.

We look forward to receiving your revised manuscript.

Kind regards,

Meijing Wang, MD

Academic Editor

PLOS ONE

Journal Requirements:

"This study was supported by a grant-in-aid from the Japan Agency for Medical Research and Development, and JSPS KAKENHI Grant Number JP20K08400, Japan,"

Please respond by return e-mail so that we can amend your financial disclosure and competing interests on your behalf.

"This study was supported by a grant-in-aid from the Japan Agency for Medical Research and Development, and JSPS KAKENHI Grant Number JP20K08400, Japan,"

"This study was supported by a grant-in-aid from the Japan Agency for Medical Research and Development, and JSPS KAKENHI Grant Number JP20K08400, Japan,"

Reviewers' comments:

Reviewer's Responses to Questions

**Comments to the Author**

1. Is the manuscript technically sound, and do the data support the conclusions?

Reviewer #1: Yes

Reviewer #2: Yes

2. Has the statistical analysis been performed appropriately and rigorously? 

Reviewer #1: Yes

Reviewer #2: Yes

3. Have the authors made all data underlying the findings in their manuscript fully available?

Reviewer #1: Yes

Reviewer #2: No

4. Is the manuscript presented in an intelligible fashion and written in standard English?

Reviewer #1: Yes

Reviewer #2: Yes

5. Review Comments to the Author

Reviewer #1: The paper submitted by Yamada Y et al. includes studies that Muse cell product treatment significantly attenuated acute myocardial infarction (AMI) injury in mini-pigs in an effective and safe way. In fact, how to achieve the safety and effectiveness is still a key issue in stem cell therapy, while, this study has made beneficial attempts. This paper is well written, and the results are interesting and beneficial to further advance stem cell preclinical experiments in vivo. The authors should concern the following defects:

1 In Fig. 1D, the percentage of the infarction area should be detected by short-axis TTC staining, and the percentage of the fibrotic area should be detected by Masson-Trichrome staining

2 Evidence should be shown for no any tumor formation in any organs in the Muse cell treatment group, for example, organs, including heart, kidney; lung; liver and pancreas, are fixed and cut into sections to stain with H-E to detect the tumor formation upon histological examination in any animals not only in the gross appearance of the organ but also under the microscope.

3 In Table1, the plasma activity of creatine kinase (CK)-MB and lactate dehydrogenase, as well as the plasma content of TNF-α should be included.

Reviewer #2: In this manuscript, the authors tested human Muse (multilineage-differentiating stress enduring) cells in a mini-pig model of acute myocardial infarction (AMI), to determine their therapeutic potential in AMI. The experimental design was straightforward. The data shown are solid and well-organized. However, a few things (listed below) the authors could have addressed better.

1. More details should be provided regarding how human Muse cells are generated. Do these Muse cells express HLA-G for better immune tolerance? What made them different from MSCs?

2. In “animal model and protocol” method section, it is not clear why some animals received 10^7 human Muse cells, while “additional AMI animals (n=3) received …10^6 cells”, for what purpose?

3. In this study, the authors used mini-pig AMI model. What’s the difference between mini-pig and adult pig model? Is this mini-pig model sufficient as a preclinical model?

4. In Fig. 2, vehicle-treated infarct border area should be included as negative controls.

5. Fig. 3 needs better organization and explanation. Fig. 3B&3C are confusing (what’s the take-home message?). Fig. 3D should be included in the text, rather than as part of a figure.

6. PLOS authors have the option to publish the peer review history of their article (what does this mean?). If published, this will include your full peer review and any attached files.

Reviewer #1: No

Reviewer #2: **Yes: **Jianyun Liu

---

## [Author Response · Author response to Decision Letter 0]

17 Feb 2022

Responses to the editor 

• Please discuss why muse cells are superior to MSCs.

• More details should be provided regarding how human Muse cells are generated. Do these Muse cells express HLA-G for better immune tolerance? What made them different from MSCs?

Thank you for valuable comments. The information how human Muse cell product was generated was newly mentioned in the Method section, page 3 the last line as follows: 

“Human Muse cell–based product was produced from human mesenchymal stem cells (MSCs) after exposing the cells to the combination stresses.” More detailed information is not available for the authors due to the trade secret. Thank you.

Other questions are answered in the Discussion, page 10, paragraph 3 as follows:

“Previous studies comparing the effect of Muse cells and MSCs in animal models including acute myocardial infarction, aortic aneurysm and partial liver transplantation suggested the superiority of Muse cells over MSCs in tissue repair effect (13, 17, 18). This was explained by several mechanisms; 1) systemically administered Muse cells selectively home to damaged site by sensing sphingosine-1-phosphate (S1P), one of the general signals of tissue injury produced by phosphorylating the cell membrane component sphingosine in damaged cells, by using S1P receptor 2. 2) After homing, Muse cells spontaneously differentiate into cells that comprise the tissue and replace damaged/dying cells, and survive as integrated cells in the host tissue for an extended period of time. 3) In contrast to Muse cells, MSCs are mainly trapped in the lung after systemic administration and they do not show specific homing to damaged site (13). Furthermore, MSCs disappear from the whole body by ~2 weeks after administration (13). Even if a small number of MSCs home to the damaged tissue, their differentiation potential is limited to osteocytes, adipocytes and chondrocytes, and are unable to differentiate into other mesodermal linages or ectodermal- or endodermal-lineages (10). Thus, replacement of damaged/dying cells may not be efficiently conducted in MSCs. 4) Muse cells have the ability to produce cytokines and trophic factors relevant to tissue protection, anti-apoptosis and anti-fibrosis at the similar level to that of MSCs. Since Muse cells selectively home to damaged site and are integrated, those by-stander effect might be long-lasting compared to MSCs (13, 17, 18). 5) High percent of Muse cells express HLA-G, known to related to immunotolerance in the placenta, compared to MSCs. This might enable Muse cells to survive in the host tissue for a longer time compared to MSCs (13). 6) Muse cells are contained in MSCs as 1~several percent of total population. Even though Muse cells are contained, the majority of non-Muse cells may mask the beneficial effect of Muse cells when MSCs are administered. One of the possible mechanisms might be competing S1P and inhibit specific homing of Muse cells to damaged tissue.”

• With respect to clinical use, are the muse cells easy to isolate and expand to a large amount in vitro?

Thank you for the comment. According to the advice, we newly inserted the sentence below in Discussion, page 10, paragraph 2 as follows:

“Muse cells can be isolated as SSEA-3(+) cells from various sources since they normally reside in the bone marrow (~0.03% of mononucleated cell population), peripheral blood and organ connective tissue (7, 20). However, practical sources will be the bone marrow, adipose tissue, dermal fibroblasts and umbilical cord. Muse cells are also collectable from commercially released MSCs and fibroblasts, and are contained as several percent of these cultured cells (7, 20). Therefore, Muse cells are accessible. 

The doubling time of Muse cells is ~1.3 days/cell division (7, 20). This is nearly the same or slightly longer than that of human fibroblasts. Since they are not tumorigenic, unlike ES and iPS cells, they do not show exponential proliferative activity. Nevertheless, they are expandable to a clinical scale. Indeed, Life Science Institute Inc., a group company of Mitsubishi Chemical Holdings Corporation, succeeded in producing clinical grade Muse cell formula, CL2020. CL2020 is already applied to clinical trials (19). “

Responses to the reviewer #1 

Reviewer #1: The paper submitted by Yamada Y et al. includes studies that Muse cell product treatment significantly attenuated acute myocardial infarction (AMI) injury in mini-pigs in an effective and safe way. In fact, how to achieve the safety and effectiveness is still a key issue in stem cell therapy, while, this study has made beneficial attempts. This paper is well written, and the results are interesting and beneficial to further advance stem cell preclinical experiments in vivo. The authors should concern the following defects:

1 In Fig. 1D, the percentage of the infarction area should be detected by short-axis TTC staining, and the percentage of the fibrotic area should be detected by Masson-Trichrome staining

Thank you for the comments. It is generally accepted that the Evans blue dye/TTC method is not reliable for evaluating infarct size 72 hours after the onset and/or reperfusion because of remodeling of the heart due to scar shrinkage within the infarct region (14). For this reason, we used Masson-Trichrome staining to assess the infarct size as a percentage of LV at 2 weeks after AMI.

We added these sentences to the Method section as follows, page 6, paragraph 1, lines 2-5:

“Since it is generally accepted that the Evans blue dye/TTC method is not reliable for evaluating infarct size 72 hours after reperfusion because of remodeling of the heart tissue due to scar shrinkage within the infarct region (14), we used Masson-Trichrome staining to assess the infarct size as the percentage of LV.”

2 Evidence should be shown for no any tumor formation in any organs in the Muse cell treatment group, for example, organs, including heart, kidney; lung; liver and pancreas, are fixed and cut into sections to stain with H-E to detect the tumor formation upon histological examination in any animals not only in the gross appearance of the organ but also under the microscope in heart, liver, lung, kidney, and spleen.

Thank you for the comment. 

We did not perform the pathological assessment whether the administration of Muse cells caused any tumor formation because the endpoint was set at 14 days in this study and it was too short time period to evaluate tumor formation. We have already examined whether intravenous administration of Muse cells caused tumor formation in a rabbit model of AMI at 2 months and 6 months after the administration (Yamada et al. Cir Res 2018). As a result, there was no tumor formation not only in the gross appearance of the organ but also under the microscope. Similarly, mouse stroke model that received human Muse cells did not show any tumor formation in the brain, liver, spleen, kidney and lung for up to 6 months (Uchida et al., Stroke 2017, 48:428-435.). 

3 In Table1, the plasma activity of creatine kinase (CK)-MB and lactate dehydrogenase, as well as the plasma content of TNF-α should be included.

Thank you for the comment.

Although we did not measure CK-MB, LDH oｒ plasma TNF-α in the present study, we confirmed in the first-in-human clinical trial that administration of Muse cell product did not affect biological test such as LDH, ALT, AST, ALP, r-GTP, total bilirubin, BUN and creatinine, and inflammatory cytokines such as IL-1�, TNF-α, IL-6 and INF-γduring 12 weeks (19). However, we agree that the point raised by the referee is very important. We added the sentence in the Discussion section of the revised version as follows, page 12, paragraph 1, lines 1- 4:

“We confirmed in this first-in-human clinical trial that administration of Muse cell product did not affect biological test such as LDH, ALT, AST, ALP, r-GTP, total bilirubin, BUN and creatinine, and inflammatory cytokines such as IL-1�, TNF-α, IL-6 and INF-γ during 12 weeks (19).”

 

Responses to the Reviewer #2

Reviewer #2: In this manuscript, the authors tested human Muse (multilineage-differentiating stress enduring) cells in a mini-pig model of acute myocardial infarction (AMI), to determine their therapeutic potential in AMI. The experimental design was straightforward. The data shown are solid and well-organized. However, a few things (listed below) the authors could have addressed better.

1. More details should be provided regarding how human Muse cells are generated. Do these Muse cells express HLA-G for better immune tolerance? What made them different from MSCs?

Thank you for valuable comments. The information how human Muse cell product was generated was newly mentioned in the Method section, page 3 the last line as follows: 

“Human Muse cell–based product was produced from human mesenchymal stem cells (MSCs) after exposing the cells to the combination stresses.” More detailed information is not available for the authors due to the trade secret. Thank you.

In regard to HLA-G and difference between Muse cells and MSCs are described in Discussion, page 10, paragraph 3:

“Previous studies comparing the effect of Muse cells and MSCs in animal models including acute myocardial infarction, aortic aneurysm and partial liver transplantation suggested the superiority of Muse cells over MSCs in tissue repair effect (13, 17, 18). This was explained by several mechanisms; 1) systemically administered Muse cells selectively home to damaged site by sensing sphingosine-1-phosphate (S1P), one of the general signals of tissue injury produced by phosphorylating the cell membrane component sphingosine in damaged cells, by using S1P receptor 2. 2) After homing, Muse cells spontaneously differentiate into cells that comprise the tissue and replace damaged/dying cells, and survive as integrated cells in the host tissue for an extended period of time. 3) In contrast to Muse cells, MSCs are mainly trapped in the lung after systemic administration and they do not show specific homing to damaged site (13). Furthermore, MSCs disappear from the whole body by ~2 weeks after administration (13). Even if a small number of MSCs home to the damaged tissue, their differentiation potential is limited to osteocytes, adipocytes and chondrocytes, and are unable to differentiate into other mesodermal linages or ectodermal- or endodermal-lineages (10). Thus, replacement of damaged/dying cells may not be efficiently conducted in MSCs. 4) Muse cells have the ability to produce cytokines and trophic factors relevant to tissue protection, anti-apoptosis and anti-fibrosis at the similar level to that of MSCs. Since Muse cells selectively home to damaged site and are integrated, those by-stander effect might be long-lasting compared to MSCs (13, 17, 18). 5) High percent of Muse cells express HLA-G, known to related to immunotolerance in the placenta, compared to MSCs. This might enable Muse cells to survive in the host tissue for a longer time compared to MSCs (13). 6) Muse cells are contained in MSCs as 1~several percent of total population. Even though Muse cells are contained, the majority of non-Muse cells may mask the beneficial effect of Muse cells when MSCs are administered. One of the possible mechanisms might be competing S1P and inhibit specific homing of Muse cells to damaged tissue. ”

• With respect to clinical use, are the muse cells easy to isolate and expand to a large amount in vitro?

Thank you for the comment. We newly mentioned about the question on page 12, paragraph 2:

“Muse cells can be isolated as SSEA-3(+) cells from various sources since they normally reside in the bone marrow (~0.03% of mononucleated cell population), peripheral blood and organ connective tissue (7, 20). However, practical sources will be the bone marrow, adipose tissue, dermal fibroblasts and umbilical cord. Muse cells are also collectable from commercially released MSCs and fibroblasts, and are contained as several percent of these cultured cells (7, 20). Therefore, Muse cells are accessible. 

 The doubling time of Muse cells is ~1.3 days/cell division (7, 20). This is nearly the same or slightly longer than that of human fibroblasts. Since they are not tumorigenic, unlike ES and iPS cells, they do not show exponential proliferative activity. Nevertheless, they are expandable to a clinical scale. Indeed, Life Science Institute Inc., a group company of Mitsubishi Chemical Holdings Corporation, succeeded in producing clinical grade Muse cell formula, CL2020. CL2020 is already applied to clinical trials (19).”

2. In “animal model and protocol” method section, it is not clear why some animals received 10^7 human Muse cells, while “additional AMI animals (n=3) received …10^6 cells”, for what purpose?

Thank you for the comment.

In the Method section, we stated that: “Animals received intravenous injection of 1x107 semi-clinical grade human Muse cell preparation (2 mL) (Muse group, n=5) or 2 mL of saline (Vehicle group, n=7) at 24 hours after reperfusion of occluded coronary artery for 30 min without immunosuppressants and observed for 2 weeks. Additional AMI animals (n=3) received intravenous injection of 1x106 of GFP-labeled-human Muse cells (7).” 

 The Muse cells administered 1 x 107 cells were semi-clinical grade human Muse cell preparation provided by Clio, Inc. (merged into Life Science Institute, Inc.Tokyo) which were not labelled with GFP. These cells were used for the assessment of effects on the infarct size and function. 

On the other hand, to examine whether human Muse cell differentiate into cardiomyocytes and vessels in mini-pig AMI model, we infused the lesser number of human Muse cells labeled with GFP cells,1x106. Therefore, the purpose was different each other. Since the point raised by the referee is indeed important and information was insufficient, we newly added the sentence below in page 5, first paragraph:

“For the assessment of effects on the infarct size reduction and function recovery, animals received intravenous injection of 1x107 semi-clinical grade human Muse cell preparation (2 mL) (Muse group, n=5) or 2 mL of saline (Vehicle group, n=7) at 24 hours after reperfusion of occluded coronary artery for 30 min without immunosuppressants and observed for 2 weeks. For the assessment of Muse cell differentiation into cardiomyocytes and vessels in histologic analysis, additional AMI animals (n=3) received intravenous injection of 1x106 of GFP-labeled-human Muse cells (13). The investigators evaluating the outcomes were blinded to the protocol.” 

3. In this study, the authors used mini-pig AMI model. What’s the difference between mini-pig and adult pig model? Is this mini-pig model sufficient as a preclinical model?

Thank you for the comment. 

Adult pigs such as Yorkshire pigs have too big weights of approximately 100-200 Kg, and mini-pigs have weights of approximately 10-20 kg. We considered that a mini-pig is sufficient as one of the preclinical models. 

4. In Fig. 2, vehicle-treated infarct border area should be included as negative controls.

Thank you for the comment.

As suggested by the reviewer, vehicle treated negative controls were added to the Figure 2 (Figure 2-B) and Figure 3 (Figure 3-B) in the revised version. 

Fig. 2B

Fig. 3B

5. Fig. 3 needs better organization and explanation. Fig. 3B&3C are confusing (what’s the take-home message?). Fig. 3D should be included in the text, rather than as part of a figure.

Thank you for the comment.

As suggested by the reviewer, we deleted Fig. 3D in the previous version. Consequentl y, Fig 3 was rearranged as follows:

Figure 3 Differentiation of Muse cells into vessels 

A: GFP-labelled Muse cells and CD31-positive microvessels in the infarct border area. 

GFP (green) and CD31 (Red) are merged, suggesting that GFP-labelled Muse cells differentiated into vascular endothelium. Bar = 50 µm

B: Negative control in the infarct border area. CD31 (red), DAPI (blue), Bar = 25 µm

C: Typical pictures of CD31-positive microvessels by immunohistological staining in the Vehicle and Muse groups. Bars = 50 µm

D: The number of CD31-positive microvessels by immunohistological staining is significantly greater in the Muse (n=5) group than in the Vehicle (n=7) group.

Because original Fig. 3B and Fig. 3 C were confusing, as pointed out by the reviewer, we explained these data in revised Figure 4 B & 4-C in the Figure Legends. Original Fig. 3B and 3C were moved to Fig 4 in the revised version. The revised legend for Fig 4 is rewrote as follows: 

On page 17, paragraph 3, lines 3-8:

“B, C: Episode summary for 2 weeks in a case of Muse groups, showing no arrythmias such as bradycardia, pause except for tachycardia. When we see the recorded electrocardiogram as shown in C, this was judged as a 300/min of tachycardia in the loop recorder. However, careful analysis of electrocardiogram revealed that the recorder judged T wave and R wave as a one beat and then the heart rate was regarded as a 300/min of tachycardia, but it was actually a 150/min of heart rate. Therefore, this was not a real tachycardia of 300/min.”

---

## [Decision Letter · Decision Letter 1]

1 Mar 2022

Human Muse cells reduce myocardial infarct size and improve cardiac function without causing arrythmias in a swine model of acute myocardial infarction

PONE-D-21-37007R1

Dear Dr. Minatoguchi,

We’re pleased to inform you that your manuscript has been judged scientifically suitable for publication and will be formally accepted for publication once it meets all outstanding technical requirements.

Kind regards,

Meijing Wang, MD

Academic Editor

PLOS ONE

Additional Editor Comments (optional):

Reviewers' comments:

Reviewer's Responses to Questions

**Comments to the Author**

1. If the authors have adequately addressed your comments raised in a previous round of review and you feel that this manuscript is now acceptable for publication, you may indicate that here to bypass the “Comments to the Author” section, enter your conflict of interest statement in the “Confidential to Editor” section, and submit your "Accept" recommendation.

Reviewer #1: All comments have been addressed

Reviewer #2: All comments have been addressed

2. Is the manuscript technically sound, and do the data support the conclusions?

Reviewer #1: Yes

Reviewer #2: Yes

3. Has the statistical analysis been performed appropriately and rigorously? 

Reviewer #1: Yes

Reviewer #2: Yes

4. Have the authors made all data underlying the findings in their manuscript fully available?

Reviewer #1: Yes

Reviewer #2: No

5. Is the manuscript presented in an intelligible fashion and written in standard English?

Reviewer #1: Yes

Reviewer #2: Yes

6. Review Comments to the Author

Reviewer #1: This manuscript was well written. The authors have answered all my questions, now I have no more qeustions.

Reviewer #2: (No Response)

7. PLOS authors have the option to publish the peer review history of their article (what does this mean?). If published, this will include your full peer review and any attached files.

Reviewer #1: No

Reviewer #2: **Yes: **Jianyun Liu

---

## [Editor Report · Acceptance letter]

9 Mar 2022

PONE-D-21-37007R1 

 Human Muse cells reduce myocardial infarct size and improve cardiac function without causing arrythmias in a swine model of acute myocardial infarction 

Dear Dr. Minatoguchi:

I'm pleased to inform you that your manuscript has been deemed suitable for publication in PLOS ONE. Congratulations! Your manuscript is now with our production department. 

Kind regards, 

on behalf of

Dr. Meijing Wang 

Academic Editor

PLOS ONE